# Colposcopic Impression Has a Key Role in the Estimation of the Risk of HSIL/CIN3

**DOI:** 10.3390/cancers13061224

**Published:** 2021-03-11

**Authors:** Marta del Pino, Martina Aida Angeles, Cristina Martí, Carla Henere, Meritxell Munmany, Lorena Marimon, Adela Saco, Natalia Rakislova, Jaume Ordi, Aureli Torné

**Affiliations:** 1Institute Clinic of Gynecology, Obstetrics, and Neonatology, Hospital Clínic, University of Barcelona, 08036 Barcelona, Spain; MARTI@clinic.cat (C.M.); carlahenere@gmail.com (C.H.); mmunmany@clinic.cat (M.M.); atorne@clinic.cat (A.T.); 2Institut d’Investigacions Biomèdiques August Pi i Sunyer (IDIBAPS), Hospital Clínic, 08036 Barcelona, Spain; 3Department of Surgical Oncology, Institut Claudius Regaud, Institut Universitaire du Cancer de Toulouse—Oncopole, 31100 Toulouse, France; AngelesFite.Martina@iuct-oncopole.fr; 4Department of Pathology, Hospital Clínic, University of Barcelona, 08036 Barcelona, Spain; lorena.marimon@isglobal.org (L.M.); masaco@clinic.cat (A.S.); jordi@clinic.cat (J.O.); 5ISGlobal, Hospital Clínic, 08036 Barcelona, Spain; rakislova@clinic.cat

**Keywords:** colposcopy, risk-based management, HSIL/CIN3

## Abstract

**Simple Summary:**

Colposcopy impression could provide valuable information for risk estimation. We aimed to analyze the value of adding colposcopy impression to screening tests for the diagnosis of high-grade squamous intraepithelial lesion/grade 3 cervical intraepithelial neoplasia (HSIL/CIN3) in 302 women referred for colposcopy due to abnormal screening results. At least 30% of the women with grade 2 colposcopy findings had HSIL/CIN3, independent of the screening test results. Among women with an HSIL Pap smear and grade 2 colposcopy findings, 53.3% had HSIL/CIN3 independently of the hrHPV genotype. The prevalence of HSIL/CIN3 in women with <HSIL Pap smear, non-HPV 16/18 infection, and normal colposcopy or with grade 1 findings was 2.9% and 8.1%, respectively. In conclusion, a colposcopic impression provides essential information to identify women at risk of HSIL/CIN3.

**Abstract:**

Recently published guidelines stratify the risk of high-grade squamous intraepithelial lesion/grade 3 cervical intraepithelial neoplasia (HSIL/CIN3) based on hrHPV detection and Pap smear results. However, colposcopic impression could also provide valuable information for risk estimation. We aimed to analyze the value of adding colposcopic impression to screening tests for the diagnosis of HSIL/CIN3 in 302 women referred for colposcopy due to an abnormal Pap smear. All women underwent hrHPV detection and genotyping (HPV 16/18 vs. non-16/18 hrHPV), Pap smear, and colposcopy with at least one biopsy. HSIL Pap smear, HPV 16/18, and grade 2 colposcopy findings increased the risk of HSIL/CIN3 in the univariate analysis but only colposcopy retained significance in the multivariate model. At least 30% of the women with grade 2 colposcopy findings had HSIL/CIN3, independent of the screening test results. Among women with an HSIL Pap smear and grade 2 colposcopy findings, 53.3% had HSIL/CIN3 independently of the hrHPV genotype. Contrarily, the prevalence of HSIL/CIN3 in women with <HSIL Pap smear, non-HPV 16/18 infection, and normal colposcopy or with grade 1 findings was 2.9% and 8.1%, respectively. In conclusion, colposcopy impression provides essential information to identify women at risk of HSIL/CIN3.

## 1. Introduction

Adequate management of women with an abnormal Pap smear and/or high-risk human papillomavirus (hrHPV) test screening results is key in the secondary prevention of cervical cancer. According to classical screening strategies based on Pap smear, all women with an abnormal screening test result are referred to colposcopy to identify the intraepithelial precursors of cervical cancer [1]. These classical strategies have been criticized as similar management is proposed for women with marked differences in the risk of precancer.

Recently published international guidelines for cervical cancer screening highlight the need for adjusting the management of women according to their individual risk of precancer [2], proposing a risk-based algorithm [3,4,5,6,7]. However, these guidelines consider only the screening test results for risk-estimation, whereas it has been clearly shown that colposcopy examination provides key information on the risk of harboring cervical cancer precursors [8]. Indeed, it has been suggested that the risk that each particular woman has of harboring a precancer can be better estimated by combining the results of the three tests (Pap smear, hrHPV testing, and colposcopy impression) [1,8]. It has also been proposed that the need for biopsy sampling can be guided by the addition of information provided by colposcopy to the results of the screening tests [1,8]. 

Another important caveat of these risk-based strategies is the cutoff for the diagnosis of precursors of cervical cancer. Although most of the available reports evaluate the prevalence of high-grade squamous intraepithelial lesion/grade 2 and higher cervical intraepithelial neoplasia (HSIL/CIN2+) as the threshold, it is widely accepted that HSIL/CIN2 has poor reproducibility [9,10] and a relatively low progression rate [11,12], thereby making this category a suboptimal end-point. Indeed, it has been claimed that HSIL/CIN3, the unquestionable precursor of cervical cancer, is a better threshold for risk assessment. However, only a few studies present specific data for HSIL/CIN3. 

The aim of this study was to evaluate the benefits of adding colposcopy to the screening tests (Pap smear, hrHPV detection with specific genotyping of HPV 16 and 18) to determine the risk of HSIL/CIN3.

## 2. Materials and Methods

### 2.1. Study Protocol and Patient Selection

This was a prospective study conducted in the Colposcopy Unit of the Hospital Clínic of Barcelona from January 2014 to September 2015. In our health region, cervical cancer screening, at that time, was routinely based on Pap smear. In the present study we included all women referred to the Colposcopy Unit due to an abnormal screening test result within the previous six months. During the first visit at our unit, a cervical sample was obtained and preserved in a methanol-based fixative (PreservCyt solution, Hologic Corp, Marlborough, MA, USA). This material was used for liquid-based cytology, hrHPV testing, and genotyping of HPV 16 and/or HPV 18 (HPV 16/18 infection). All the women underwent digital colposcopy with a thorough examination of the lower genital tract, and one to four biopsy samples from the cervix were obtained. 

The exclusion criteria were: (1) previous treatment for HSIL/CIN2+, (2) ongoing pregnancy, and (3) synchronic vaginal intraepithelial lesions. 

### 2.2. Colposcopy and Biopsy Sampling

Colposcopy examinations were performed by six experienced colposcopists, all of whom are accredited by the Spanish Society of Cervical Pathology and Colposcopy using an Olympus EvisExera II CV-180 (Tokyo, Japan). First, 5% acetic acid was applied. In order to detect “fast fader” lesions, the acetic acid was repeatedly reapplied to the cervix using cotton balls during 1 to 2 min. Colposcopy impressions were described following the classification of the International Federation of Cervical Pathology and Colposcopy, IFCPC) [13,14]. The colposcopy was considered adequate when the cervix could be completely assessed and was not obscured by inflammation, bleeding, or scarring. The squamo-columnar junction was considered to be completely visible when 360° of the junction could be visualized (type 1 and 2 transformation zone), whereas patients for whom the squamo-columnar junction could not be completely assessed were classified as having a type 3 transformation zone. Original squamous epithelium, columnar epithelium, and the transformation zone were considered normal colposcopy findings. Smooth surface with an irregular outer border, slight acetowhite staining, slow to appear, and quick to disappear; mild, speckled partial iodine positivity, and fine punctuation or fine regular mosaic were considered as grade 1 findings suggestive of low-grade cervical intraepithelial lesions (LSIL/CIN1). Grade 2 findings suggestive of HSIL/CIN2+ were considered in the presence of a smooth surface with a sharp outer border; dense acetowhite staining appearing early and being slow to resolve; iodine negativity in a previously densely white epithelium; coarse punctuation or wide irregular mosaic; and dense acetowhite staining of the columnar epithelium [15]. 

During the colposcopy procedure, one to four colposcopy-directed biopsies from any abnormal areas or from different regions in one large complex abnormal area of the cervix were obtained. A non-targeted (random) biopsy from apparently normal epithelium was also taken within the transformation zone in patients showing normal colposcopy [16]. Thus, at least one biopsy was obtained from all patients. Additionally, endocervical curettage using a Kevorkian curette was performed in all women with a non-completely visible transformation zone and in cases in which the lesion was partially or totally visualized in the endocervix.

### 2.3. Liquid-Based Cytology and hrHPV Testing

Cervical samples were collected with a cytobrush and stored in PreservCyt solution (Hologic Corp, Marlborough, MA, USA) for ThinPrep liquid-based cytology and hrHPV testing. We used the Thin-Prep T2000 slide processor (Hologic) to prepare thin-layer cytology slides, and these were stained using the Papanicolaou method. A cytotechnologist evaluated the cytology slides, and then a pathologist confirmed the results using the revised Bethesda nomenclature [17] 

For hrHPV testing and genotyping, the Cobas HPV test (Cobas 4800; Roche Molecular Diagnostics), based on a real-time polymerase chain reaction (PCR) system, was used. This method detects 14 high-risk HPV types and provides specific information on HPV 16/18 infection (limited genotyping).

### 2.4. Histological Diagnosis

All the histological samples were fixed in 10% neutral buffered formalin and embedded in paraffin following routine procedures. Four µm sections were stained with hematoxylin and eosin. All the histological samples were reviewed to confirm or exclude the presence or absence of SIL/CIN and its grade. In all the cervical samples obtained, p16 immunohistochemical staining (CINtec histology kit, clone E6H4; mtm-Roche Laboratories, Heidelberg, Germany) was performed [15]. The histological diagnosis was based on hematoxylin and eosin criteria, however p16 block staining was required to perform a diagnosis of HSIL/CIN2+ [18]. Biopsy specimens were classified as normal, LSIL/CIN1, HSIL/CIN2, and HSIL/CIN3 according to the latest WHO classifications [19]. 

### 2.5. Statistical Analysis

Data analysis was performed using SPSS version 25.0 software (SPSS, Inc., Chicago, IL, USA). The final diagnostic categories were defined as follows: (1) negative for SIL/CIN included women with a negative biopsy, negative hrHPV test, and Pap smear showing either normal smear or atypical squamous cell of uncertain significance (ASC-US); (2) LSIL/CIN1 included: (a) all women with a histological diagnosis of LSIL/CIN1, independent of the results of hrHPV testing and Pap smear, (b) women with a negative biopsy and negative hrHPV test, and LSIL Pap smear, and (c) women with a negative biopsy and a positive hrHPV test, independent of the cytological result; and (3) HSIL/CIN2+ included all women with a histological diagnosis of HSIL/CIN2, HSIL/CIN3, or adenocarcinoma in situ, independent of the results of the Pap smear and hrHPV testing. Since risk estimates for HSIL/CIN3 are much less heterogeneous than HSIL/CIN2+ results [10,20,21], women with HSIL/CIN3 (including patients with a histological diagnosis of HSIL/CIN3 or adenocarcinoma in situ) were also segregated in the analysis and considered as the main end-point of the study. 

For risk estimation, “high-risk” results included either a HSIL result in Pap smear, and/or HPV 16/18 infection, and/or grade 2 findings in the colposcopy impression. “Low-risk” included a <HSIL Pap smear result, a negative result for HPV 16/18, and colposcopy not showing grade 2 findings. 

Categorical variables are presented as absolute numbers and percentages and compared using the Chi-square or Fisher’s exact test. Continuous variables are presented as mean and standard deviation (SD) and compared using the Student’s T-test or Wilcoxon test. Independent prognostic factors for HSIL/CIN2+ and HSIL/CIN3 were evaluated using univariate and multivariate logistic regression models. *p* values < 0.05 were considered statistically significant.

Since patients with a type 3 transformation zone in whom grade 2 findings were not identified may harbor non-visible endocervical HSIL/CIN3, and thus, the colposcopic impression may underestimate the prevalence of precancer, calculations were performed primarily excluding these women, although the results of the overall group (including women with a type 3 transformation zone and normal colposcopy or with grade 1 findings) are also presented to show the impact on the accuracy that these cases might have in the estimation of risk. 

## 3. Results

### 3.1. Pap Smear, hrHPV Testing, Colposcopic Impression, and Final Diagnosis 

Three hundred and two women fulfilled the inclusion criteria and were included in the study. The mean age, demographic information (immunological status and smoking habits), and referral Pap smear are shown in Table 1. 

The results of the Pap smear performed during the first visit at the Colposcopy Unit were negative in 23 women (7.6%), showing ASC-US in 24 (8.0%) women, LSIL in 139 (46.0%) women, and HSIL in 116 (38.4%) women. The correlation between the referral Pap smear and the Pap smear result in the first visit is shown in Table 2. 

Sixty-eight women (22.5%) had a negative hrHPV test result, 108 (35.8%) had infection by a non-16/18 hrHPV, and 126 (41.6%) presented with HPV 16/18 infection (111 women showed HPV 16 infection, 21 HPV 18 infection, among whom, 6 had an infection by HPV 16 and 18). Overall, colposcopy showed normal findings in 116 women (38.4%), abnormal grade 1 findings in 88 (29.1%), and abnormal grade 2 findings in 98 (32.4%) women. Fifty-two of the 116 (44.8%) women with normal colposcopy findings and 19 of the 88 (21.6%) women with grade 1 colposcopy findings showed a type 3 transformation zone. Among the 98 women with grade 2 findings, 40 (40.8%) had a type 3 transformation zone.

The final histological diagnosis was negative for SIL/CIN in 128 women (42.4%), showing LSIL/CIN1 in 51 (16.9%) and HSIL/CIN2+ in 123 patients (40.7%, 42 HSIL/CIN2 and 81 HSIL/CIN3). The results of the Pap smear, the hrHPV with 16/18 genotyping, and the colposcopic impression in the different final histological diagnostic categories are shown in Table 3. Similar results were obtained when including the referral Pap test instead of the Pap test performed in the first visit in the analysis (data not shown). Among women with HPV 16/18, HSIL/CIN3 was found in 33.3% (35/105) of those with a single HPV 16 infection, in 26.7% (4/15) of the women with a single HPV 18 infection, and in 50.0% (3/6) of the women with HPV 16 and HPV 18 co-infection (*p* = 0.700). No differences found between HPV 16 and HPV 18 in terms of the prevalence of HSIL/CIN2+ (data not shown).

### 3.2. Correlation between the Combined Results of the Tests (Pap Smear, hrHPV Testing, and Colposcopy) and the Final Diagnosis

Figure 1 shows the prevalence of HSIL/CIN3 and HSIL/CIN2+ according to the combined results of the tests in the 231 women in whom the absence of grade 2 colposcopy findings could be confidently excluded (women with a type 1 or 2 transformation zone and women with a type 3 transformation zone with grade 2 findings). There was a correlation between the results of the tests and the risk of harboring histological HSIL/CIN3: among the 60 women with an HSIL Pap smear result and grade 2 colposcopy findings, 53.3% had HSIL/CIN3. In this subset of women, hrHPV genotyping (HPV 16/18 vs. non-16/18 hrHPV infection) did not modify the risk of HSIL/CIN3. Interestingly, women with grade 2 findings in the colposcopy showed a risk of HSIL/CIN3 greater than or equal to 30%, even in those with a screening test not suggestive of HSIL/CIN (i.e., women with a LSIL Pap smear and a negative hrHPV test). In contrast, the risk of HSIL/CIN3 among women with the “low-risk” combination of results was less than 10%. Remarkably, the correlation for the combined results of the tests and the prevalence of HSIL/CIN2+ was much lower. Sixty-four of the 116 women with a normal colposcopy (55.2%) had a completely visible transformation zone. The prevalence of HSIL/CIN3 in these women was 7.8% (5/64). Among these women, 34 showed low-risk screening results (<HSIL cytology and non-HPV 16/18 infection). The prevalence of HSIL/CIN3 in these women was 2.9% (1/34). Among the 69 women with grade 1 colposcopy findings and a visible transformation zone, 13 (18.8%) showed HSIL/CIN3. Thirty-seven of these women showed neither HSIL nor HPV 16/18 infection. The prevalence of HSIL/CIN3 in these women was 8.1% (3/37). Globally, among the 18 women with HSIL/CIN3+ and normal colposcopy or grade 1 colposcopy findings showing a completely visible transformation zone, 10 (55.6%) had HPV 16/18 infection and 7 (38.9%) HSIL Pap smear results.

Figure 2 shows the prevalence of HSIL/CIN3 and HSIL/CIN2+ according to the results of the tests in the overall group of 302 women included in the study. The correlation with the results of the tests was clearly lower. Among the 52 women with normal colposcopy and a type 3 transformation zone, 8 (15.4%) had HSIL/CIN3 and 11 (21.2%) HSIL/CIN2+. Among the 19 women with grade 1 colposcopy findings and a type 3 transformation zone, 6 (31.6%) had HSIL/CIN3 and 10 (52.6%) HSIL/CIN2+. The diagnosis of all these cases was made in the endocervical curettage. 

Table 4 shows the univariate and multivariate analyses for the final diagnosis of HSIL/CIN3 and HSIL/CIN2+. In the univariate analysis, an HSIL Pap smear result, HPV 16/18 infection, and grade 2 colposcopy findings significantly increased the risk of HSIL/CIN3. However, only grade 2 colposcopy findings remained significant in the multivariate analysis (odds ratio (OR): 5.4, 95% confidence interval (CI): 2.6–11.5; *p* < 0.001). Similar results were obtained when including the referral Pap test instead of the Pap test performed in the first visit in the analysis (data not shown). Similar results were also obtained after excluding patients with a type 3 transformation zone and normal colposcopy or showing grade 1 findings, in which an underlying HSIL/CIN3 lesion in the endocervix cannot be discarded by colposcopy (data not shown). 

Table 5 shows the sensitivity and specificity of the different screening test results for the detection of HSIL/CIN2+ and HSIL/CIN3+. As a single test, a Pap test result of HSIL and hrHPV testing positive for HPV 16/18 had the highest sensitivity for the detection of HSIL/CIN3 (63.0%; 95% CI: 44.2–78.5; and 63.0%; 95% CI: 52.1–72.7, respectively), whereas grade 2 colposcopy findings had the highest specificity (77.8%; 95% CI: 71.9–82.8). A Pap test result of HSIL, and HPV 16/18 infection, and grade 2 colposcopy findings had a very high specificity for the diagnosis of HSIL/CIN3 (90.5%; 95% CI 85.9–93.7), albeit with a low sensitivity (30.8%, 95% CI 21.6–41.7).

## 4. Discussion

In the present series, an HSIL result in the Pap smear, HPV 16/18 infection, and grade 2 colposcopy findings were significantly associated with underlying HSIL/CIN3 (and HSIL/CIN2+). However, colposcopic impression showed the strongest association, and in the multivariate analysis it was the only result associated with the final histological diagnosis of HSIL/CIN3, supporting the inclusion of a colposcopic impression together with screening tests to effectively stratify the risk of HSIL/CIN3 in women undergoing cervical cancer screening [3,4]. Indeed, in our series, grade 2 findings in the colposcopy evaluation increased the specificity for the diagnosis of HSIL/CIN3 from 81.0% to 90.5% in women with an HSIL Pap smear and HPV 16/18 infection.

Recent cervical cancer screening strategies propose a risk-based management approach that considers the results of the screening tests (Pap smear and/or hrHPV limited genotyping) [4,5]. However, it has been shown that colposcopy also provides relevant information on the risk of HSIL/CIN3, as clearly highlighted in this study [8,22]. Indeed, in the present series the percentage of women with HSIL/CIN3 increased with the severity of colposcopy abnormalities, and in women with grade 2 colposcopy findings, this percentage was higher than 30%, even among women showing “low-risk” screening test results (<HSIL Pap smear result and a negative HPV test or non-16/18 hrHPV infection). These results are in keeping with previous reports suggesting that risk-estimation should include the three tests (Pap smear, HPV 16/18 testing, and colposcopic impression) [1] and that women with two or more “high-risk” results (HSIL in the Pap smear, HPV 16/18 infection, and grade 2 colposcopy findings) should be considered as being at “high-risk” due to the very high prevalence of underlying HSIL/CIN3 [7,8]. 

Recent American guidelines recommend immediate treatment (without histological confirmation) for women with an HSIL Pap smear result and HPV 16/18 infection, on the basis of an estimated risk of HSIL/CIN3 greater than 60% [8]. Despite our results confirming the high prevalence of HSIL/CIN2+ in these patients, the prevalence of HSIL/CIN3 was much lower, especially in women with normal colposcopy or with grade 1 findings when their transformation zone was completely visible (11.1% and 33.3%, respectively). Thus, colposcopy evaluation with thorough biopsy sampling is probably a better strategy than immediate treatment for these patients to avoid an unacceptable rate of overtreatment [23]. This means neither increasing the number of women referred to colposcopy nor considering colposcopy as part of the routine screening, but rather taking into consideration the information provided by the biopsy.

In contrast, women showing low-risk results in the three tests had a very low risk of HSIL/CIN3. The prevalence of HSIL/CIN3 in women with a Pap smear result <HSIL, no HPV 16/18 infection, and a visible transformation zone was 2.9% and 8.1%, for women with normal colposcopy and grade 1 findings, respectively. The low prevalence of HSIL/CIN3 in the low-risk patients observed in this series indicates that biopsies are probably unnecessary in these women, as previous reports have suggested [1,8,22].

This study has several valuable strengths. Firstly, all women were prospectively recruited, and all cases were very well characterized with liquid-based cytology, hrHPV testing with genotyping of HPV 16 and 18, and colposcopy examination with thorough biopsy sampling performed at the same time to obtain precise risk estimation. This protocol offers high confidence of not missing a prevalent lesion, allowing accurate test correlation and the correct evaluation of the risk of premalignant cervical lesions. Moreover, we show not only the risk of HSIL/CIN2+ but also the risk of HSIL/CIN3, the unquestionable precursor of cervical cancer. Finally, all risk-based guidelines have been based on the analysis of American cohorts, whereas it is well known that the risk may vary in different geographical areas [24,25]. Our study provides data from Spain, which may help to refine the risk-based management algorithms for European settings.

Our study also has some limitations. Firstly, all women were referred due to an abnormal Pap smear, as this was the standard cervical cancer screening test used in our country. This approach does not adequately evaluate the usefulness of hrHPV testing, since women with hrHPV infection and a normal Pap smear are not included. All the women included in this study had a previous abnormal Pap smear, and therefore represent a selected subgroup of women at high-risk of underlying premalignant lesions. Another limitation is the relatively small number of patients included, which resulted in a low number of women in some combinations of test results. However, although the relatively small sample size might overestimate the prevalence of high-grade lesions in some combinations, the percentage of HSIL/CIN3 observed in groups including more than 10 women are in keeping with the rates observed in previous reports [1,8,22]. 

## 5. Conclusions

In conclusion, the results of our study highlight the essential role of colposcopy in identifying women with abnormal screening test results at high risk for underlying HSIL/CIN3. The addition of colposcopic impression can refine the management of women with abnormal screening test results to avoid missing HSIL/CIN3 lesions without overtreating an unacceptable number of patients.

## Figures and Tables

**Figure 1 cancers-13-01224-f001:**
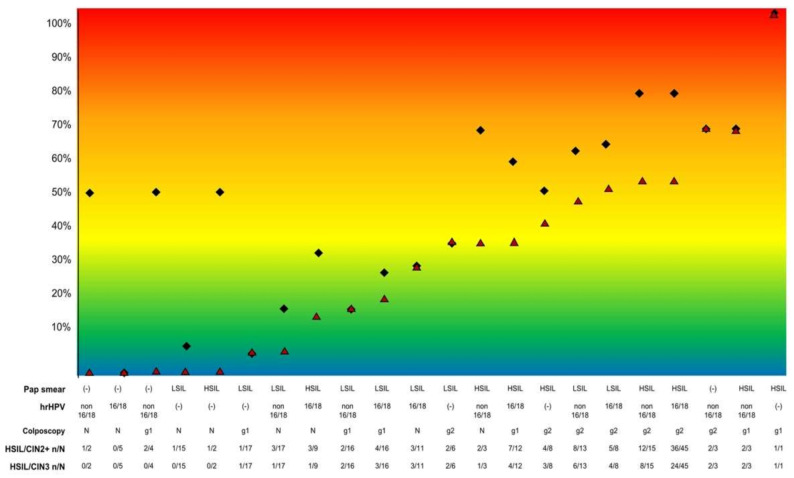
Prevalence of HSIL/CIN2+ (black diamonds) and HSIL/CIN3 (red triangles) according to the Pap test result, hrHPV testing, and colposcopic impression including the 231 women in whom the presence or absence of grade 2 colposcopy lesions could be confidently excluded or confirmed (all women with a type 1 or 2 transformation zone and women with a type 3 transformation zone with grade 2 findings). No women showed a negative Pap-test with HPV 16/18 infection or grade 1 or grade 2 findings in the colposcopy evaluation. Footnote: (−): negative; LSIL: low-grade squamous intraepithelial neoplasia (including atypical squamous cell of undetermined significance (ASC-US)); HSIL: high-grade squamous intraepithelial neoplasia; hrHPV: high-risk human papillomavirus; HSIL/CIN2+: high-grade squamous intraepithelial lesion/grade 2 or 3 cervical intraepithelial neoplasia or adenocarcinoma in situ. HSIL/CIN3: high-grade squamous intraepithelial lesion/grade 3 cervical intraepithelial neoplasia or adenocarcinoma in situ. N: normal colposcopy; g1: grade 1 findings; g2: grade 2 findings.

**Figure 2 cancers-13-01224-f002:**
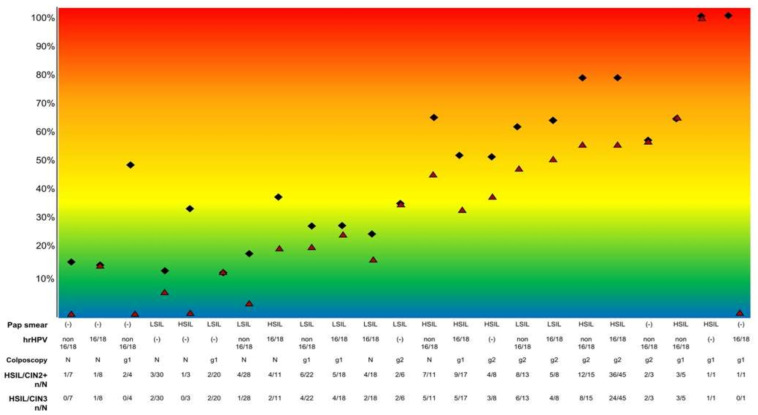
Prevalence of HSIL/CIN2+ (black diamonds) and HSIL/CIN3 (red triangles) according to the Pap test result, hrHPV testing and genotyping, and colposcopic impression considering the 231 women in whom the presence or absence of grade 2 colposcopy lesions could be confidently excluded or confirmed (all women with a type 1 or 2 transformation zone and women with a type 3 transformation zone with grade 2 findings). No women showed a negative Pap test with HPV 16/18 infection or grade 2 findings at the colposcopy evaluation. Footnote: (−): negative; LSIL: low-grade squamous intraepithelial neoplasia (including atypical squamous cell of undetermined significance (ASC-US)); HSIL: high-grade squamous intraepithelial neoplasia; hrHPV: high-risk human papillomavirus; HSIL/CIN2+: high-grade squamous intraepithelial lesion/grade 2 or 3 cervical intraepithelial neoplasia or adenocarcinoma in situ. HSIL/CIN3: high-grade squamous intraepithelial lesion/grade 3 cervical intraepithelial neoplasia or adenocarcinoma in situ. N: normal colposcopy; g1: grade 1 findings; g2: grade 2 findings.

**Table 1 cancers-13-01224-t001:** Epidemiological information (age, immunological status, smoking habits, and referral cytology) of the 302 women included in the study.

Patients Characteristics	*n*	(%)
Age (mean)	37.6	±10.3
Immunosuppression		
No	296	(98.0%)
Yes	6	(2.0%)
Smoking habits		
No	151	(50.0%)
Current smoker	40	(13.2%)
Yes*	111	(36.8%)
*Cigarettes per day		
<10	58	(52.3%)
10–20	43	(38.7%)
>20	10	(9.0%)
Referral Pap smear		
ASC-US	53	(17.5%)
LSIL	120	(39.8%)
HSIL	129	(42.7%)

Number of cigarretes per day considering only former smokers.

**Table 2 cancers-13-01224-t002:** Correlation between the Pap smear result of the referral visit and the Pap smear result of the first visit performed at the Colposcopy Unit.

Referral Pap Smear	Pap Smear Result of the First Visit							
	Negative		ASC-US		LSIL		HSIL	
ASC-US	10	(18.9)	7	(13.2)	25	(47.2)	11	(20.7)
LSIL	7	(5.8)	11	(9.2)	77	(64.2)	25	(20.8)
HSIL	6	(4.6)	6	(4.6)	37	(28.7)	80	(62.1)

ASC-US: atypical squamous cells of uncertain significance; LSIL: low-grade squamous intraepithelial lesion; HSIL: high-grade squamous intraepithelial lesion (includes atypical glandular cell results (AGC) and ASC cannot exclude HSIL (ASC-H)).

**Table 3 cancers-13-01224-t003:** Correlation between the Pap test result of first visit, hrHPV status and genotyping (negative vs. non-16/18 hrHPV vs. HPV 16/18), colposcopic impression, and the final diagnosis. Values are presented as absolute numbers and percentages.

First Visit Tests Results		Final Histological Diagnosis								
	*n*	Negative (*n* = 128)		LSIL/CIN1 (*n* = 51)		HSIL/CIN2 (*n* = 42)		HSIL/CIN3 (*n* = 81)		*p*
Pap smear result of first visit										
Negative	23	13	(56.6)	3	(13.0)	4	(17.4)	3	(13.0)	<0.001
ASC-US	24	14	(58.4)	5	(20.8)	3	(12.5)	2	(8.3)	
LSIL	139	76	(54.7)	29	(20.8)	9	(6.5)	25	(18.0)	
HSIL	116	25	(21.5)	14	(12.1)	26	(22.4)	51	(44.0)	
hrHPV result										
Negative	68	48	(70.6)	7	(10.3)	3	(4.4)	10	(14.7)	<0.001
Non-16/18 hrHPV	108	44	(40.7)	19	(17.6)	16	(14.8)	29	(26.9)	
HPV 16/18	126	36	(28.6)	25	(19.8)	23	(18.3)	42	(33.3)	
Colposcopic impression										
Normal	116	76	(65.5)	15	(12.9)	12	(10.4)	13	(11.2)	<0.001
Grade 1 findings	88	35	(39.7)	24	(27.3)	10	(11.4)	19	(21.6)	
Grade 2 findings	98	17	(17.4)	12	(12.2)	20	(20.4)	49	(50.0)	

ASC-US: atypical squamous cells of uncertain significance; LSIL: low-grade squamous intraepithelial lesion; HSIL: high-grade squamous intraepithelial lesion; HSIL/CIN2+: high-grade squamous intraepithelial lesion/grade 2–3 cervical intraepithelial neoplasia or adenocarcinoma in situ; HSIL/CIN3: high-grade squamous intraepithelial lesion/grade 3 cervical intraepithelial neoplasia or adenocarcinoma in situ; hrHPV: high-risk human papillomavirus.

**Table 4 cancers-13-01224-t004:** Univariate and multivariate logistical regression analysis of risk factors associated with the diagnosis of high-grade cervical squamous intraepithelial lesions or worse HSIL/CIN2+ and HSIL/CIN3.

Risk Factors	HSIL/CIN3						HSIL/CIN2+					
	Univariate Analysis			Multivariate Analysis			Univariate Analysis			Multivariate Analysis		
	OR	(95% CI)	*p*	OR	(95% CI)	*p*	OR	(95% CI)	*p*	OR	(95% CI)	*p*
Age												
≤35 years	1						1					
>35 years	0.9	(0.6–1.5)	0.773			NA	1.1	(0.7–1.7)	0.804			NA
Pap smear of the first visit												
Negative	1			1			1			1		
LSIL	1.3	(0.4–4.8)	0.668			NA	0.7	(0.3–1.9)	0.500			NA
HSIL	5.2	(1.5–18.6)	0.011	2.0	(1.2–3.5)	0.011	4.5	(1.7–11.9)	0.002	2.7	(0.9–7.9)	0.064
hrHPV genotyping												
Negative	1			1			1			1		
Non-16/18 hrHPV	2.1	(0.9–4.7)	0.062	1.7	(0.7–4.0)	0.208	3.0	(1.5–6.2)	0.002	2.4	(1.1–5.3)	0.036
HPV 16/18	2.9	(1.3–6.2)	0.006	2.0	(0.9–4.8)	0.099	4.5	(2.2–9.1)	<0.001	2.7	(1.2–6.1)	0.015
Colposcopic impression												
Normal	1			1			1			1		
Grade 1 findings	2.2	(1.0–4.7)	0.047	2.0	(0.9–4.5)	0.074	1.8	(1.0–3.3)	0.069	1.8	(0.9–3.4)	0.098
Grade 2 findings	7.9	(3.9–15.9)	<0.001	5.4	(2.6–11.5)	<0.001	9.0	(4.8–16.7)	<0.001	5.5	(2.8–10.9)	<0.001

OR: odds ratio; CI: confidence interval; NA: not applicable. LSIL: low-grade squamous intraepithelial lesion; HSIL: high-grade squamous intraepithelial lesion; HSIL/CIN2+: high-grade squamous intraepithelial lesion/grade 2–3 cervical intraepithelial neoplasia or adenocarcinoma in situ; HSIL/CIN3: high-grade squamous intraepithelial lesion/grade 3 cervical intraepithelial neoplasia or adenocarcinoma in situ; hrHPV: high-risk human papillomavirus.

**Table 5 cancers-13-01224-t005:** Sensitivity and specificity of the HSIL cytology, HPV 16 and or 18 infection, abnormal findings grade 2 at the colposcopic impression and combination of the three results for the diagnosis of HSIL/CIN2+ and HSIL/CIN3.

Test Result	HSIL/CIN2+				HSIL/CIN3			
	Sensitivity	(95% CI)	Specificity	(95% CI)	Sensitivity	(95% CI)	Specificity	(95% CI)
Single test results								
HSIL Pap	62.6	(53.8–70.6)	78.2	(71.6–83.6)	63.0	(44.2–78.5)	70.6	(64.3–83.6)
HPV 16/18	52.8	(44.1–61.4)	65.9	(58.7–72.5)	63.0	(52.1–72.7)	70.6	(64.3–76.2)
Grade 2 findings	56.1	(47.3–64.5)	84.3	(78.2–88.9)	60.5	(49.6–70.4)	77.8	(71.9–82.8)
Combination of two tests								
HSIL Pap + HPV 16/18	39.8	(31.6–48.7)	86.6	(80.8–90.8)	38.3	(28.4–49.2)	81.0	(75.3–85.2)
HSIL Pap + grade 2 findings	41.8	(33.4–50.7)	91.1	(86.0–94.4)	43.2	(33.0–54.1)	85.1	(79.8–89.2)
HPV 16/18 + grade 2 findings	33.3	(25.6–42.1)	93.3	(88.6–96.1)	34.6	(25.1–45.4)	88.7	(83.8–92.1)
Combination of the three tests								
HSIL Pap + HPV 16/18 + grade 2 findings	29.3	(21.9–37.8)	95.0	(90.7–97.3)	30.8	(21.6–41.7)	90.5	(85.9–93.7)

HSIL: high-grade squamous intraepithelial lesion; HSIL/CIN2+: high-grade squamous intraepithelial lesion/grade 2–3 cervical intraepithelial neoplasia or adenocarcinoma in situ; HSIL/CIN3: high-grade squamous intraepithelial lesion/grade 3 cervical intraepithelial neoplasia or adenocarcinoma in situ; HPV 16/18: HPV genotype 16 and/or 18.

## Data Availability

Subject numbers will be assigned sequentially to women enrolled in the study. All data collected in this research protocol will be treated as confidential and will be identified with the woman’s study number and not with the woman’s name or address. The cervical cytology results and the biopsy and LEEP results will be recorded in the registry of histopathology and cytopathology as well as in the patients’ clinical file. A safety physician safeguards the investigational code. All study records will be archived in a safe and secure location. To verify the accuracy of the data, these records will be available to the representatives of the national government (e.g., Inspection of Public Health) and licensed inspectors of foreign governments. Members of the Medical Ethical committee are allowed to inspect the quality of the accomplished research. All study data and human materials will be kept for 15 years.

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
