# Peer review of "Colposcopic Impression Has a Key Role in the Estimation of the Risk of HSIL/CIN3"

_cancers, 2021, doi:10.3390/cancers13061224_

Round 1

Reviewer 1 Report

This is a very interesting manuscript regarding the use of colposcopy impression as an additional key parameter for HSIL/CIN3+ risk stratification. The results support the case that women with two or more "high risk" results (HSIL on cytology, HPV 16/18 infection and grade 2 colposcopy findings) should be considered "high risk" due to the very high prevalence of underlying HSIL/CIN3. They also suggest that HSIL cytology/HPV1618 positive women with normal/grade1 colposcopy findings would benefit from thorough biopsy sampling rather than immediate treatment as recommended by the recent ASCCP guidelines.

- Why is the referral cytology result not used instead of the cytology result of the first visit? I understand that there is a mismatch between the two tests. But if I understood the methods correctly, at the time of the colposcopy (first visit) the colposcopist only had the result of the referral cytology. Therefore decisions/risk stratification are taken upon the referral result, not the second one. I would add an analysis in which only this cytological result is taken into account. Additionally, one could consider adding the HPV test result, assuming full concordance between the referral cytology and the first visit test.

- The figures are very informative and illustrative. However, I wonder if the authors could add a table/figure (or modify existing ones) with the results of specificity, sensitivity, PPV, NPV, for each single test and their combinations in a cumulative way, to see how the tests effectively stratify the risk. Another option is to show it as a decision tree. 
Citology 
Citology + HPV 16/18
Citology + HPV 16/18 + colposcopic impression 

- Please also add 95% Confidence intervals to Figures 1 and 2. Risks in categories with very low N can be misleading.

- Were differences found between HPV16 and HPV18 positive cases?

Minor comments

- Figures 1 and 2. Is it possible to sort results first by cytology result, then by HPV status and colposcopy impression 
- line 228: the paragraph is misplaced. It is part of the footnote of Figure 1

Author Response

REVIEWER 1

QUESTION 1: Why is the referral cytology result not used instead of the cytology result of the first visit? I understand that there is a mismatch between the two tests. But if I understood the methods correctly, at the time of the colposcopy (first visit) the colposcopist only had the result of the referral cytology. Therefore decisions/risk stratification are taken upon the referral result, not the second one. I would add an analysis in which only this cytological result is taken into account. Additionally, one could consider adding the HPV test result, assuming full concordance between the referral cytology and the first visit test.

RESPONSE 1: We understand the concern raised by the reviewer. Indeed, at the time of the first visit the colposcopists only had the result of the referral cytology. It is also true that, as shown in table 2, there was a significant mismatch between the referral cytology and the new cytology taken at the first visit. However, the risk evaluation analysis provided similar results independently of whether the referral cytology or the cytology at first visit was used in the analysis. In the next two tables we show the results of these analyses using the referral cytology.

Final histological diagnosis

n

Negative

(n= 128)

LSIL/CIN1

(n= 51)

HSIL/CIN2

(n= 42)

HSIL/CIN3

(n= 81)

p

Referral Pap smear

ASC-US

53

33

(62.3)

4

(7.5)

8

(15.1)

8

(15.1)

<0.001

LSIL

120

67

(55.8)

28

(23.3)

10

(8.3)

15

(12.5)

HSIL

129

28

(21.7)

19

(14.7)

28

(18.6)

58

(45.0)

Pap smear of the first visit

Negative

23

13

(56.6)

3

(13.0)

4

(17.4)

3

(13.0)

<0.001

ASC-US

24

14

(58.4)

5

(20.8)

3

(12.5)

2

(8.3)

LSIL

139

76

(54.7)

29

(20.8)

9

(6.5)

25

(18.0)

HSIL

116

25

(21.5)

14

(12.1)

26

(22.4)

51

(44.0)

HSIL/CIN3

HSIL/CIN2+

Univariate analysis

Univariate analysis

OR

(95% CI)

p

OR

(95% CI)

P

Referral Pap smear

Negative

1

1

LSIL

0.8

(0.3-2.0)

0.664

0.6

(0.3-1.3)

0.184

HSIL

4.5

(2.0-10.5)

0.001

4.0

(2.0-8.0)

0.001

Pap smear of the first visit

Negative

1

1

LSIL

1.3

(0.4-4.8)

0.668

0.7

(0.3-1.9)

0.500

HSIL

5.2

(1.5-18.6)

0.011

4.5

(1.7-11.9)

0.002

Thus, we finally decided to use the results of the cytology at first visit in order to provide a better picture of patient status at the time point of the initial evaluation. In this regard, it should be emphasized that the time-lapse between referral cytology and the colposcopy evaluation was variable from woman to woman. In the revised version, two new sentences have been included in the Results section (page 6, lines 222-223 in the revised version, underlined the newly added sentence), stating that the use of the referral cytology in the analysis would have resulted in similar results. We have not, however, added the previously presented tables:

“The results of the Pap smear, the hrHPV with 16/18 genotyping and the colposcopy impression in the different final histological diagnostic categories are shown in Table 3. Similar results were obtained when including the referral Pap test instead of the Pap test performed in the first visit in the analysis (data not shown). “

Also, a new sentence (underlined text) has been added on page 8, lines 293-294 of the revised version.

Table 4 shows the univariate and multivariate analyses for the final diagnosis of HSIL/CIN3 and HSIL/CIN2+. In the univariate analysis, an HSIL Pap smear result, HPV 16/18 infection and grade 2 colposcopy findings significantly increased the risk of HSIL/CIN3. However, only grade 2 colposcopy findings remained significant in the multivariate analysis (odds ratio [OR]: 5.4, 95% confidence interval [CI]: 2.6-11.5; p<0.001). Similar results were obtained when including the referral Pap test instead of the Pap test performed in the first visit in the analysis (data not shown). Similar results were also obtained after excluding patients with a type 3 transformation zone and normal colposcopy or showing grade 1 findings, in which an underlying HSIL/CIN3 lesion in the endocervix cannot be discarded by colposcopy (data not shown).

QUESTION 2:  The figures are very informative and illustrative. However, I wonder if the authors could add a table/figure (or modify existing ones) with the results of specificity, sensitivity, PPV, NPV, for each single test and their combinations in a cumulative way, to see how the tests effectively stratify the risk. Another option is to show it as a decision tree.

Citology

Citology + HPV 16/18

Citology + HPV 16/18 + colposcopic impression

RESPONSE 2: Following the reviewer’s suggestion we have added a new table (Table 5) with the results of sensitivity and specificity for each single test and their combinations in a cumulative way to show how the tests stratify the risk. The addition of the table has also resulted in an additional paragraph in the Results section (page 9, lines 305-312 in the revised version):

“Table 5 shows the sensitivity and specificity of the different screening test results for the detection of HSIL/CIN2+ and HSIL/CIN3+. As a single test, a Pap test result of HSIL and hrHPV testing positive for HPV16/18 had the highest sensitivity for the detection of HSIL/CIN3 (63.0%; 95%CI: 44.2-78.5; and 63.0%; 95%CI: 52.1-72.7, respectively), whereas grade 2 colposcopy findings had the highest specificity (77.8%; 95%CI: 71.9-82.8). A Pap test result of HSIL, and HPV16/18 infection and grade 2 colposcopy findings had a very high specificity for the diagnosis of HSIL/CIN3 (90.5; 95%CI 85.9-93.7), albeit with a low specificity (30.8%, 95%CI 21.6-41.7).”

Also, an additional sentence has been added to the Discussion section (page 10, lines 326-328 in the revised version, underlined the new sentence):

Indeed, in our series, grade 2 findings in the colposcopy evaluation increased the specificity for the diagnosis of HSIL/CIN3 from 81.0% to 90.5% in women with an HSIL Pap smear and HPV16/18 infection.

QUESTION 3:  Please also add 95% Confidence intervals to Figures 1 and 2. Risks in categories with very low N can be misleading.

RESPONSE 3: Figure 1 and 2 do not show risk. As stated in the manuscript and the figure legends, they show the prevalence of HSIL/CIN3 and HSIL/CIN2+ according to the combined results of the tests in the women in whom the absence of grade 2 colposcopy findings could be confidently excluded. For this reason, the 95% confidence intervals are not included. We agree that the numbers are small in some categories, but the figures present the whole picture of the cases included in the study, grouped according to the results of the screening tests, which as commented in the previous question by the reviewer, is vey illustrative.

QUESTION 4: Were differences found between HPV16 and HPV18 positive cases?  

RESPONSE 4: Although the current strategies proposed for cervical cancer screening do not differentiate between HPV16 and HPV18 infection, we agree with the reviewer that it could be interesting to look for possible differences between the two types in terms of risk of HSIL/CIN3. In our study, there were no differences in terms of risk of progression to HSIL/CIN3 between HPV16 and HPV18. A new sentence has been added to the Results section (page 5, lines 212-213, underlined the newly added text) to clarify this issue:

Sixty-eight women (22.5%) had a negative hrHPV test result, 108 (35.8%) had infection by a non 16/18 hrHPV, and 126 (41.6%) presented HPV16/18 infection (111 women showed HPV 16 infection, 21 HPV 18 infection, among whom, 6 had an infection by HPV 16 and 18).

And a new paragraph has been also added to this section (page 6, lines 223-226, underlined the new paragraph):

The results of the Pap smear, the hrHPV with 16/18 genotyping and the colposcopy impression in the different final histological diagnostic categories are shown in Table 3. There were no statistically significant differences if we consider the referral Pap test or the Pap test performed in the first visit related to the final histological results (data not shown). Among women with HPV 16/18, HSIL/CIN3 was found in 33.3% (35/105) of those with a single HPV 16 infection, 26.7% (4/15) of the women with a single HPV 18 infection and 50.0% (3/6) of the women with HPV 16 and HPV 18 co-infection (p=0.700). Neither were any differences found between HPV 16 and HPV 18 in terms of the prevalence of HSIL/CIN2+ (data not shown).

MINOR COMMENTS

QUESTION 5:  Figures 1 and 2. Is it possible to sort results first by cytology result, then by HPV status and colposcopy impression

RESPONSE 5: All the tests were performed in the same moment, since we used the cytology performed in the first visit and not the referral cytology (see question 1 and response 1). Thus, we did not consider cytology as the starting test. We think that it is more illustrative to show the results in a way that the reader can appreciate the increase in risk of HSIL/CIN3, which is the main outcome of the study. Following this rationale, we have sorted the results according to the prevalence of HSIL/CIN3.

QUESTION 6: line 228: the paragraph is misplaced. It is part of the footnote of Figure 1

QUESTION 6: Following the reviewer’s observation, we have corrected the error by adding the paragraph to the footnote of Figure 1. In the revised version, the cited footnote is now as follows (underlined the paragraph placed in the correct position) (page 7, lines 264-269):

Figure 1. Prevalence of HSIL/CIN2+ (black diamonds) and HSIL/CIN3 (red triangles) according to the Pap test result, hrHPV testing and colposcopy impression including the 231 women in whom the presence or absence of grade 2 colposcopy lesions could be confidently excluded or confirmed (all women with a type 1 or 2 transformation zone and women with a type 3 transformation zone with grade 2 findings).  No women showed a negative Pap-test with HPV 16/18 infection or grade 1 or grade 2 findings in the colposcopy evaluation. Footnote: (-): negative; LSIL: low-grade squamous intraepithelial neoplasia (including atypical squamous cell of undetermined significance [ASC-US]); HSIL: high-grade squamous intraepithelial neoplasia; hrHPV: high-risk Human papillomavirus; HSIL/CIN2+: high grade squamous intraepithelial lesion/ grade 2 or 3 cervical intraepithelial neoplasia or adenocarcinoma in situ. HSIL/CIN3: high grade squamous intraepithelial lesion/ grade 3 cervical intraepithelial neoplasia or adenocarcinoma in situ. N: normal colposcopy; g1: grade 1 findings; g2: grade 2 findings.

Reviewer 2 Report

Thank you for allowing me to review this paper.  IT was very well written and had some interesting concepts. 

1. There should be a Table 1 with basic demographic information.

2. The point of a screening test is ease of use and for it to be cost-effective.  Adding in colposcopy to routine screening would cause more women to undergo an invasive procedure that is likely not required in most of them.  It is thought that we frequently already perform colposcopy too frequently.  This should be included in the discussion as a weakness.

3. The argument would better be made to not perform see-and-treat (HSIL pap--> excisional procedure), but instead perform colposcopy as the second step after an abnormal pap.  However, it should not be part of routine screening.  If reworded, I think this would be a more acceptable concept.  

Author Response

REVIEWER 2

QUESTION 1: There should be a Table 1 with basic demographic information.

RESPONSE 1: Following the reviewer’s recommendation a new table with the basic demographic information has been added to the manuscript (Table 1). Also, a new sentence has been added to the Results section (page 4, lines 170-172, underlined the new sentence and crossed out the deleted words)

Three hundred two women fulfilled the inclusion criteria and were included in the study. The mean age, demographic information (immunological status and smoking habits) and referral Pap smear are shown in table 1. The mean age of the patients was 37.6 (SD ± 10.3). Fifty-three women (17.5%) were referred due to a Pap smear result of ASC-US; 120 (39.8%) due to LSIL, and 129 (42.7%) due to a HSIL cytology.

As a result, former tables 1, 2, and 3, have become tables 2, 3 and 4 in the revised version.

QUESTION 2: The point of a screening test is ease of use and for it to be cost-effective.  Adding in colposcopy to routine screening would cause more women to undergo an invasive procedure that is likely not required in most of them.  It is thought that we frequently already perform colposcopy too frequently.  This should be included in the discussion as a weakness.

RESPONSE 2: All the women included in the present study had already been referred to colposcopy according to current guidelines of the cervical cancer screening program, i.e. there was no increase in the number of women who underwent colposcopy. Indeed, the contribution of our study is that the inclusion of colposcopy information in the risk evaluation better tailors the management of women referred to colposcopy by avoiding overtreating women who not require conization, as already stated in the manuscript (page 10, lines 349-351 and page 11, lines 383-387).

The modulation of the number of women referred to colposcopy is a prior step not analyzed in this study.

QUESTION 3: The argument would better be made to not perform see-and-treat (HSIL pap--> excisional procedure), but instead perform colposcopy as the second step after an abnormal pap.  However, it should not be part of routine screening. If reworded, I think this would be a more acceptable concept. 

RESPONSE 3: We agree with the reviewer that the main message of our findings is that a careful colposcopy evaluation of patients with abnormal screening tests should be performed in order to better tailor the management of these women. As previously stated (see reply to question 2) this means neither increasing the number of women referred to colposcopy nor considering colposcopy as part of the routine screening. As suggested by the reviewer, we have reworded a sentence in the discussion to emphasize the inadequateness of see and treat strategies. (page 10, lines 351-353, underlined the new words).

Thus, colposcopy evaluation with thorough biopsy sampling is probably a better strategy than immediate treatment for these patients to avoid an unacceptable rate of overtreatment [23]. This means neither increasing the number of women referred to colposcopy nor considering colposcopy as part of the routine screening but rather taking into consideration the information provided by the biopsy.

Reviewer 3 Report

This paper addresses the value of colposcopy in addition to cytology and HPV testing. The data are nicely analysed and discussed, the limitations and strenghts of the study described. The manuscript is certainly welcome by everyone working in colposcopy.

2 comments: It would be of interest to learn more about the CIN 3 lesions with normal or grade one colposcopy: Is for these lesions the HPV test or the cytology more sensitive? Is the size of these CIN 3 lesions so small that they might be overseen and potentially be removed by biopsy?

This leads to the second question: Is the final LEEP histology of the HSIL patients available, did the final histology of surgery confirm the biopsy diagnosis.

Author Response

REVIEWER 3

QUESTION 1: It would be of interest to learn more about the CIN 3 lesions with normal or grade one colposcopy: Is for these lesions the HPV test or the cytology more sensitive? Is the size of these CIN 3 lesions so small that they might be overseen and potentially be removed by biopsy?

RESPONSE 1: We have added a new table (table 5 in the revised manuscript, see answer to question xx reviewer 1) specifying the sensitivity and specificity of each single test (HPV16/18, Pap smear and colposcopy) and their combinations for the detection of HSIL/CIN2+ and HSIL/CIN3 to provide an overall view of the accuracy of the different tests. However, given the low number of women with HSIL/CIN3 lesions who had a completely visible transformation zone and normal colposcopy or with grade 1 findings, analysis of the sensitivity and specificity of HPV testing or cytology could be misleading.

In the Results section we have also added a new sentence describing the HPV genotype and Pap smear results of the 18 women with normal colposcopy or colposcopy grade 1 findings and final diagnosis of HSIL/CIN3 to stress that issue, which was also already shown in Figure 1 and 2 (page 6-7, lines 355-358, underlined the new sentence)

Sixty-four of the 116 women with a normal colposcopy (55.2%) had a completely visible transformation zone. The prevalence of HSIL/CIN3 in these women was 7.8% (5/64). Among these women, 34 showed low-risk screening results (<HSIL cytology and non HPV 16/18 infection). The prevalence of HSIL/CIN3 in these women was 2.9% (1/34). Among the 69 women with grade 1 colposcopy findings and a visible transformation zone, 13 (18.8%) showed HSIL/CIN3. Thirty-seven of these women showed neither HSIL nor HPV 16/18 infection. The prevalence of HSIL/CIN3 in these women was 8.1% (3/37). Globally, among the 18 women with HSIL/CIN3+ and normal colposcopy or grade 1 colposcopy findings showing a completely visible transformation zone, 10 (55.6%) had HPV16/18 infection and 7 (38.9%) HSIL Pap smear result.

Unfortunately, we do not have the lesion size of the 13 HSIL/CIN3+ lesions with grade 1 colposcopy findings and a completely visible transformation zone (for HSIL/CIN3+ with normal colposcopy, there is no colposcopically measurable lesion size). Although the hypothesis suggested by the reviewer is completely reasonable, either the follow-up of these women or the conization specimen would be necessary to confirm the removal of the lesion. Since our study provides a transversal picture of the prevalence of HSIL/CIN3+, the outcome of the follow-up was not considered, despite its undoubtable interest.

QUESTION 2: Is the final LEEP histology of the HSIL patients available, did the final histology of surgery confirm the biopsy diagnosis?

RESPONSE 2: Following national guidelines, not all women with HSIL/CIN2+ lesions underwent excisional treatment. Women younger than 35 years old with a completely visible transformation zone and small lesions (less than 50% of the cervical surface) were followed every six months for a maximum of two years. A few years ago our group published a couple of studies showing that women fulfilling these criteria had a high probability of regression of the lesion (see Munmany M et al. BJOG 2017 and Munmany M et all. J Low Genit Tract Dis. 2018.) as previously suggested by the reviewer. In the same line, women with HSIL/CIN2+ and negative HPV testing frequently do not show lesion in the LEEP specimen, as previously shown by our group (Mandredi A et al. Gynecol Oncol 2013). Thus, women diagnosed with HSIL/CIN2+ are carefully selected for treatment or follow-up based on these criteria. As a result, we do not have the result of the LEEP in all women included in the study. In addition, it has been shown that the concordance between the histological diagnosis of the directed biopsy and the LEEP specimen in women with small lesions or negative HPV testing would not be perfect in many cases, as suggested by the reviewer and shown in our previously published studies.

Reviewer 4 Report

Although this work reports similar results to previous studies, I agree with the findigs of the study. It would have been very interesting to evaluate the effectiveness of colposcopy in identifying any dysplasia in patients with a positive HPV test and a negative Pap test

Author Response

REVIEWER 4

QUESTION 1: It would have been very interesting to evaluate the effectiveness of colposcopy in identifying any dysplasia in patients with a positive HPV test and a negative Pap test

RESPONSE 1: We agree with the reviewer that the value of the colposcopy can be analysed not only for the detection of HSIL/CIN3 (or HIS/CIN 2+) but also for the detection dysplasia of any grade. However, because the goal of the screening strategies is the detection of true premalignant lesions at risk of progression to cancer, we focused on HSIL/CIN2+ and HSIL/CIN3 HSIL/CIN3. The results of the colposcopy evaluation of the three women with a negative Pap smear and HSIL/CIN3 (two with a completely visible transformation zone at the colposcopy evaluation) and the four women with a negative Pap smear and HSIL/CIN2 (three with a completely visible transformation zone at the colposcopy evaluation) are shown in Figure 1 and Figure 2. However, the number of cases is too small to draw any conclusion on the value of colposcopy in this specific subset of women.

However, in order to provide the accuracy of the colposcopy, alone or in combination with the screening tests (HPV16/18 and Pap smear) for the detection of HIS/CIN2+ and SIL/CIN3, a new table (table 5 in the revised manuscript, see answer to question xx reviewer 1) has been added to the manuscript presenting the sensitivity and specificity for each single test and their combinations in a cumulative way.
